# Altered Calcium Influx Pathways in Cancer-Associated Fibroblasts

**DOI:** 10.3390/biomedicines9060680

**Published:** 2021-06-16

**Authors:** Francisco Sadras, Teneale A. Stewart, Mélanie Robitaille, Amelia A. Peters, Priyakshi Kalita-de Croft, Patsy S. Soon, Jodi M. Saunus, Sunil R. Lakhani, Sarah J. Roberts-Thomson, Gregory R. Monteith

**Affiliations:** 1School of Pharmacy, The University of Queensland, Brisbane, QLD 4102, Australia; f.sadras@uq.edu.au (F.S.); m.robitaille@uq.edu.au (M.R.); a.peters1@uq.edu.au (A.A.P.); sarahrt@uq.edu.au (S.J.R.-T.); 2Mater Research, Translational Research Institute, The University of Queensland, Brisbane, QLD 4102, Australia; teneale@me.com; 3Centre for Clinical Research, Faculty of Medicine, The University of Queensland, Herston, QLD 4006, Australia; p.kalita@uq.edu.au (P.K.-d.C.); j.saunus@uq.edu.au (J.M.S.); s.lakhani@uq.edu.au (S.R.L.); 4South Western Sydney Clinical School, University of New South Wales, Liverpool, NSW 2170, Australia; p.soon@unsw.edu.au; 5Department of Surgery, Bankstown Hospital, Bankstown, NSW 2200, Australia; 6Medical Oncology Group, Ingham Institute for Applied Medical Research, Liverpool, NSW 2170, Australia; 7Pathology Queensland, The Royal Brisbane and Women’s Hospital, Herston, QLD 4029, Australia

**Keywords:** breast cancer, cancer-associated fibroblasts, calcium, calcium signalling, voltage-gated calcium channels, store-operated calcium entry

## Abstract

Cancer-associated fibroblasts (CAFs) represent an important component of the tumour microenvironment and are implicated in disease progression. Two outstanding questions in cancer biology are how CAFs arise and how they might be targeted therapeutically. The calcium signal also has an important role in tumorigenesis. To date, the role of calcium signalling pathways in the induction of the CAF phenotype remains unexplored. A CAF model was generated through exogenous transforming growth factor beta 1 (TGFβ1) stimulation of the normal human mammary fibroblast cell line, HMF3S (HMF3S-CAF), and changes in calcium signalling were investigated. Functional changes in HMF3S-CAF calcium signalling pathways were assessed using a fluorescent indicator, gene expression, gene-silencing and pharmacological approaches. HMF3S-CAF cells demonstrated functionally altered calcium influx pathways with reduced store-operated calcium entry. In support of a calcium signalling switch, two voltage-gated calcium channel (VGCC) family members, Ca_V_1.2 and Ca_V_3.2, were upregulated in HMF3S-CAFs and a subset of patient-derived breast CAFs. Both siRNA-mediated silencing and pharmacological inhibition of Ca_V_1.2 or Ca_V_3.2 significantly impaired CAF activation in HMF3S cells. Our findings show that VGCCs contribute to TGFβ1-mediated induction of HMF3S-CAF cells and both transcriptional interference and pharmacological antagonism of Ca_V_1.2 and Ca_V_3.2 inhibit CAF induction. This suggests a potential therapeutic role for targeting calcium signalling in breast CAFs.

## 1. Introduction

It is increasingly appreciated that the tumour microenvironment (TME) is an active player in carcinogenesis [1]. Fibroblasts are the major cell type of the breast TME and are highly plastic [2]. Normal fibroblasts (NFs) are typically tumour-suppressive but can be transformed into tumour-promoting cancer-associated fibroblasts (CAFs) through a series of phenotypic changes [3]. CAFs are a highly diverse population that can also be derived from adipocytes, immune, and endothelial cells. In all of these scenarios there are significant cellular and gene expression changes [4]. A variety of CAF inducers such as TGFβ1 [5] and complement signalling [6] have been discovered; however, the molecular mechanisms behind CAF activation are not fully understood, which makes targeting CAFs a difficult endeavour. One largely unexplored mechanism, potentially involved in CAF activation, is Ca^2+^ signalling.

Ca^2+^ signalling is a critical pathway in various cellular phenotypic transitions, suggesting that it also may be involved in the transformation of NFs into CAFs. One transformation that is Ca^2+^ signalling-dependent is epithelial to mesenchymal transition (EMT). EMT is a phenomenon where epithelial cancer cells adopt a more invasive and therapy-resistant phenotype [7,8]. Various Ca^2+^ pumps are remodelled during this process, highlighting a role for Ca^2+^ signalling in malignant disease progression and in cellular remodelling [9,10]. Changes in cytosolic free calcium ([Ca^2+^]_CYT_) also play a key role in a diverse range of cellular functions including cell migration and proliferation [11] as well as cellular phenotypic changes [10,12].

Two calcium signalling pathways that are remodelled in both in vivo cancer and in vitro cancer models are store-operated calcium entry (SOCE) and Ca^2+^ influx via voltage-gated calcium channels (VGCCs) [13,14]. However, these have not been extensively evaluated in the context of CAFs. SOCE is the predominant calcium influx pathway in non-excitable cells [15,16]. In contrast, VGCCs are predominantly expressed in excitatory cells [17] but their overexpression occurs in a variety of pathologies including cancer [14,18]. Disruption of Ca^2+^ signalling by aberrant expression or activity of these two groups of calcium channels is well established in various cancer types [11], such as the role of Orai1 in breast cancer metastasis [13] and T-type Ca^2+^ channels in breast cancer cell proliferation [19]. Additionally, a switch from VGCC-mediated Ca^2+^ influx to a SOCE-dependent pathway occurs in the context of balloon-injured rat and mouse carotid arterial cells, indicating that these pathways may be linked [20,21,22]. Despite the identification of Ca^2+^ influx pathways in cellular transformation such as EMT in breast cancer [10], the role of Ca^2+^ influx via specific Ca^2+^ channels in CAF induction in breast cancer has not been evaluated.

Here, we report that calcium influx via the SOCE pathway is significantly reduced following induction of a CAF phenotype in the human normal breast fibroblast cell line—HMF3S [23]. We also show that two VGCC family members, Ca_V_1.2 and Ca_V_3.2, are upregulated in our in vitro CAF model as well as in a subset of patient-derived breast CAFs. Silencing of either Ca_V_1.2 or Ca_V_3.2 through siRNA or pharmacological inhibition significantly impaired CAF activation as assessed by induction of α-smooth muscle actin (αSMA). These studies highlight a potential role for VGCC-mediated calcium signalling in the induction of breast CAFs. Targeting VGCC and SOCE calcium influx pathways may represent novel pharmacological opportunities to maintain fibroblasts in the tumour-suppressive NF phenotype [24].

## 2. Materials and Methods

### 2.1. Cell Culture

Primary cultures of patient-derived NFs and CAFs were established as previously described [25]. The HMF3S human breast fibroblast cell line [23] was a generous gift from the laboratory of Prof. Michael O’Hare and Prof. Parmjit Jat (Ludwig Institute for Cancer Research, London, UK). Cells were routinely cultured in DMEM with high glucose (D6546; Sigma-Aldrich, St Louis, MO, USA) supplemented with 10% foetal bovine serum (FBS; In Vitro Technologies, Melbourne, Australia,) and 4 mM L-glutamine (25030; Life Technologies Australia, Scoresby, Australia). This media contains 1.8 mM Ca^2+^ in line with physiological conditions [11,26] and other CAF differentiation studies [5,27]. Cells were maintained at 37 °C and 5% CO_2_ in a humidified incubator and passaged every two to three days when cells were 70–90% confluent. Experiments were performed using cells between passage 3 and 8. Routine testing for mycoplasma contamination was carried out every 6 months using the MycoAlert™ Mycoplasma Detection Kit (Lonza, Basel, Switzerland) and cell lines authenticated by Short Tandem Repeat (STR) profiling using GenePrint 10 system (Promega, Madison, WI, USA) at QIMR Berghofer Medical Research Institute, Brisbane, Australia.

For TGFβ1 activation, cells were plated in 24-well tissue culture treated plates at 1.5 × 10^4^ cells per well or in 96-well plates at 6 × 10^3^ cells per well and allowed to adhere for 48 h. Cell monolayers at ~60% confluence were then washed once with serum-free media and incubated with serum-reduced media (SRM, 0.5% FBS) for 12–14 h prior to stimulation with 0.1–10 ng/mL TGFβ1 (10 µg/mL stock prepared in 10 mM citric acid and 0.1% BSA; PHG9204; Life Technologies, Scoresby, Australia) in SRM. TGFβ1-containing media was renewed daily for two days.

For pharmacological inhibitor studies, cells were incubated with SRM for 12–14 h before replacing with fresh media supplemented with ML218 (3–10 nM; SML0385; Sigma-Aldrich, St. Louis, MO, USA) or nimodipine (3–10 nM; N149; Sigma-Aldrich, St. Louis, MO, USA) and TGFβ1 (10 ng/mL) or appropriate controls.

### 2.2. siRNA Transfection

For siRNA-mediated RNA interference, cells were transfected with 100 nM Dharmacon-ONTARGETplus SMARTpool siRNAs (CACNA1C L-006123-00, CACNA1H L-006128-00, non-targeting control D-001810-10-05; Thermo Fisher Scientific, Waltham, MA, USA) 24 h after plating as previously described [28], 24 h after siRNA treatment cells were incubated with SRM overnight followed by stimulation with TGFβ1 as described above.

### 2.3. Real-Time Quantitative RT-PCR

RNA was isolated from cells using a Qiagen RNeasy Plus Mini Kit (Qiagen, Hilden, Germany) and cDNA was generated using an Omniscript RT Kit (Qiagen, Hilden, Germany). Real-time RT-PCR was performed using TaqMan^®^ Fast Universal Master Mix (Thermo Fisher Scientific, Waltham, MA, USA) and TaqMan^®^ gene expression assays using a StepOnePlus™ instrument (Applied Biosystems, Carlsbad, CA, USA) under universal cycling conditions. For assay details, see Appendix A. Levels of mRNA were normalised to 18S rRNA (patient samples) or GAPDH (HMF3S cells) and fold change was calculated using the comparative C_T_ method as previously described [29]. Mean C_T_ greater than 34 was considered as outside the limits of instrument detection and classified as ‘not detected’. For patient samples, where target mRNA was undetected in one of the two pairs a fold change of 100 was assigned.

### 2.4. Immunostaining

HMF3S cells were fixed in 4% paraformaldehyde for 15 min, washed 3× with PBS then permeabilised and blocked in blocking buffer (PBS containing 10% goat serum, 0.3 M glycine, 1% BSA, and 0.1% Triton-X-100) for 1 h at room temperature. αSMA Alexa 488-conjugated antibody (ab184675; Abcam, Cambridge, UK) was diluted 1:200 in blocking buffer and incubated overnight at 4 °C. Cells were then washed with PBS and incubated for 10 min with 1 µg/mL of DAPI in PBS at room temperature. Cells were then washed with PBS and imaged using an ImageXpress Micro imaging system with a 10× objective (Molecular Devices, Sunnyvale, CA, USA). Controls were prepared by omitting primary or secondary antibody addition (data not shown).

### 2.5. Immunoblotting

Cell lysis was performed using protein lysis buffer (Tris base 50 mM, NaCl 100 mM, IGEPAL NP-40 1%, sodium deoxycholate 0.5%) supplemented with protease and phosphatase inhibitors (Roche Applied Science, Penzberg, Germany). Gel electrophoresis was carried out using Mini-PROTEAN^®^ TGX Stain-Free™ Precast Gels (BioRad, Hercules, CA, USA) and transferred to a PVDF membrane (BioRad, Hercules, CA, USA). αSMA antibody (ab7817; Abcam, Cambridge, United Kingdom, 1:200–1:1500 in 5% skim milk powder (SMP) or 5% BSA) and incubated for 16–40 h at 4 °C with shaking based on antibody batch. Membranes were washed and incubated in goat-anti-mouse HRP-conjugated secondary antibody (170-6516; BioRad, Hercules, CA, USA, 1:10,000, 5% SMP) for 1 h at room temperature. β-Actin (A5441; Sigma-Aldrich, St. Louis, MO, USA, 1:10,000, 5% SMP) was used as a loading control on a parallel gel. PVDF membranes were incubated in SuperSignal West Dura Extended Duration Chemiluminescent Substrate (Thermo Fisher Scientific, Waltham, MA, USA) and imaged using a BioRad ChemiDoc Imaging System. Quantification of protein bands was performed using Image Lab software (version 5.2.1; BioRad, Hercules, CA, USA) as per user guidelines [30].

### 2.6. Measurement of Intracellular Free Calcium

HMF3S cells were plated at 2.5 × 10^3^ cells per well in black-walled 96-well microplates (Corning Costar, Corning, NY, USA) and treated with 0.1–10 ng/mL TGFβ1 as described above. Store-operated calcium entry (SOCE) assay was performed by loading cells with the cell permeant Ca^2+^ indicator Fluo-4 AM (4 μM; Life Technologies, Carlsbad, CA, USA) in SRM. Cells were then incubated at 37 °C for 30 min and media was subsequently replaced with physiological salt solution (PSS; 5.9 mM KCl, 1.4 mM MgCl_2_, 10 mM HEPES, 1.2 mM NaH_2_PO_4_, 5 mM NaHCO_3_, 140 mM NaCl, 11.5 mM glucose, 1.8 mM CaCl_2_) and incubated at room temperature for 15 min, protected from light. SOCE was assessed as previously described [31], with minor modifications. Briefly, cells were washed twice with PSS containing 0.3% bovine serum albumin (BSA; A3803, Sigma-Aldrich, St. Louis, MO, USA), and once with CaCl_2_-free PSS (PSS nominal Ca^2+^). PSS nominal Ca^2+^ was then added to each well. SOCE was activated by sequentially treating cells with BAPTA (100 μM, Invitrogen, Waltham, MA, USA), the sarco/endoplasmic Ca^2+^ ATPase inhibitor and the Ca^2+^ store depleting agent cyclopiazonic acid (CPA; 10 μM, Sigma-Aldrich, St. Louis, MO, USA) and CaCl_2_ (0.6 mM). Cytosolic Ca^2+^ influx was assessed using a Fluorometric Imaging Plate Reader (FLIPR^TETRA^; Molecular Devices, San Jose, CA, USA) as previously described [32]. Area under the curve (AUC) for ‘Peak 1’ (representative of store emptying) was calculated between 26 and 600 s, and ‘Peak 2’ (representative of Ca^2+^ influx) between 720 and 1400 s was used to compare SOCE, similar to previous studies [31].

### 2.7. Statistical Analysis

All statistical analyses were performed using GraphPad Prism (version 3.8.1. GraphPad Software Inc., La Jolla, CA, USA). Details of statistical analyses are provided in figure legends.

## 3. Results

### 3.1. TGFβ1 Stimulation Induces a CAF-Like Phenotype in HMF3S Cells

Given that TGFβ1 is a key TME signalling molecule [33] and well-characterised CAF inducer [5,34], we explored its ability to transform normal (non-activated) HMF3S fibroblast cells [23] into a CAF-like phenotype (HMF3S-CAF). We assessed CAF activation by measuring the transcript levels of four reported CAF markers [35]: fibroblast activation protein-α (FAPα), platelet-derived growth factor receptor-β (PDGFRβ), fibronectin and tenascin-C (Figure 1a), and protein expression of the widely used CAF marker, α-smooth muscle actin (αSMA) (Figure 1b,c) [36]. As described by Costa et al. [33] homogenous expression of αSMA was not observed by immunofluorescence, indicating a heterogeneous population of CAFs. However, given the concentration-dependent increase in all CAF markers assessed, with a significant upregulation at 1 and 10 ng/mL TGFβ1, we defined TGFβ1 at these concentrations as suitable for inducing a CAF-like phenotype in this model.

### 3.2. TGFβ1-Mediated CAF Induction Impairs SOCE in HMF3S Cells

As SOCE is the predominant Ca^2+^ influx pathway in non-excitable cells [37,38] and is remodelled in a variety of cancer types and cellular transformations [10,11,36], we sought to determine whether SOCE was altered in HMF3S cells following induction of a CAF-like phenotype. To assess this, we initiated SOCE by first depleting the endoplasmic reticulum (ER) Ca^2+^ stores using cyclopiazonic acid (CPA) to inhibit the sarco/endoplasmic reticulum Ca^2+^ ATPase (SERCA) under conditions of extracellular Ca^2+^ chelation with 1,2-bis(o-aminophenoxy)ethane-N,N,N′,N′-tetraacetic acid (BAPTA). Following the initial depletion of the ER Ca^2+^ stores, SOCE was assessed by addition of extracellular Ca^2+^. The initial CPA-mediated increase in [Ca^2+^]_CYT_, representing store emptying, did not differ between HMF3S-CAF and HMF3S cells, as assessed by the area under the curve (AUC) of peak 1 (Figure 2a,b) suggesting that CAF activation does not result in a change in ER store capacity. However, SOCE was significantly reduced at both 1 and 10 ng/mL TGFβ1 treatment as measured by the [Ca^2+^]_CYT_ increase following Ca^2+^ re-addition (AUC of peak 2) (Figure 2c) and the ratio of Ca^2+^ influx to Ca^2+^ store depletion (AUC peak 2 to peak 1) (Figure 2d). These results indicate remodelling of SOCE following TGFβ1-mediated induction of a CAF-like phenotype in HMF3S cells.

### 3.3. HMF3S Activation Remodels Calcium Channel and Channel Regulator Expression

A potential explanation for the observed decrease in SOCE in HMF3S-CAFs is altered expression of SOCE components. To determine if this was a contributing factor, we assessed mRNA levels of the SOCE Ca^2+^ channel isoforms, Orai1 and their canonical activators stromal interaction molecule 1 and 2 (STIM1 and STIM2) [37]. TGFβ1 treatment at 1 and 10 ng/mL modestly, but significantly, increased Orai1 and downregulated STIM2 mRNA levels (Figure 3a) while at 0.01 ng/mL STIM1 levels were significantly increased. This suggests that the functional decrease in SOCE is unlikely to be associated with decreased mRNA levels of the canonical SOCE components Orai1 and STIM1.

Given reports of phenotypic switching between SOCE and VGCC Ca^2+^ influx pathways in certain pathological models [21], we compared mRNA levels of all 10 VGCCs in TGFβ1-stimulated HMF3S-CAFs. Ca_V_1.1, Ca_V_1.3, Ca_V_1.4, Ca_V_2.2, Ca_V_2.3, and Ca_V_3.3 transcript levels were not detected in HMF3S or HMF3S-CAF cells (data not shown). However, mRNA levels of the L-type Ca_V_1.2 and T-type Ca_V_3.2 VGCCs were significantly upregulated following treatment with 1 and 10 ng/mL TGFβ1 (Figure 3b). Of the P/Q, N and R type VGCCs assessed, only Ca_V_2.1 was expressed; however, no significant changes in transcript levels were detected following TGFβ1 stimulation (Figure 3b). These data suggest a potential SOCE to VGCC calcium influx phenotypic switch in HMF3S-CAFs and that this is associated with increases in L- and T-type calcium channel expression.

To evaluate whether this observation was mirrored in a clinical context, we assessed 17 paired CAF samples from human breast cancer patients. Consistent with the in vitro model, there was no consistent change in Orai1, STIM1, or STIM2 transcript levels in CAFs and NFs in paired breast cancer patient samples (Figure 3c). There was, however, a much higher degree of variability in VGCC transcript levels, with a trend towards upregulation for Ca_V_1.2 and Ca_V_3.2 in CAFs compared to their paired NFs (Figure 3c).

### 3.4. Voltage-Gated Calcium Channels Are Regulators of CAF Activation in HMF3S Cells

Given the observed remodelling of T- and L-type Ca^2+^ channels in patient-derived breast CAFs as well as our in vitro model, we assessed the potential role of these calcium channels in HMF3S-CAF activation using both siRNA-mediated silencing and pharmacological inhibition. Ca_V_1.2 and Ca_V_3.2 siRNA silencing (Appendix A) significantly reduced TGFβ1-stimulated increases in αSMA by ~50% (Figure 4a). We also assessed the effects of siCa_V_1.2 and siCa_V_3.2 on mRNA transcript levels of four CAF markers which were all significantly increased by TGFβ1 (Figure 4b). Both siRNA treatments abolished the significant increases in FAPα, fibronectin, and tenascin-C mRNA induced by TGFβ1. However, such an effect was only seen with siCa_V_1.2 for the marker PDGFRβ (Figure 4b). Combined, these data suggest that Ca_V_1.2 and Ca_V_3.2 play a role in induction of the HMF3S-CAF phenotype in this model.

To evaluate the potential for commercially available therapeutics to inhibit CAF induction, we co-treated HMF3S cells with TGFβ1 and nimodipine [39], an L-type VGCC inhibitor (Ca_V_1.2 family), or ML218 [40], a T-type VGCC inhibitor (Ca_V_3.2 family). Similar to our findings using siRNA targeting of Ca_V_1.2 and Ca_V_3.2, treatment of TGFβ1-stimulated HMF3S with either nimodipine or ML218 reduced levels of αSMA by ~50% (Figure 5a,b).

## 4. Discussion

Cellular plasticity is common in developmental physiological and pathological processes, including neuronal development [41], T-cell differentiation [42], and EMT [10]. In all of these cases, calcium signalling plays important roles [10,36,39]. The plasticity of both tumour cells and cells of the TME is well established and the myriad of potential transformations possible in this context are of increasing interest [43]. Despite the established links between Ca^2+^ signalling and phenotypic changes in cancer cells, our understanding of Ca^2+^ signalling during CAF transformation remains limited. However, studies linking Ca^2+^ signalling and CAFs are emerging. Recently, Vancauwenberghe et al. [44] showed that overexpression of a mutant isoform of the Ca^2+^ permeable ion channel transient receptor potential cation channel subfamily A member 1 (TRPA1) in CAFs promotes resistance to resveratrol-mediated apoptosis in prostate cancer cells, and that this occurs in a calcium-dependent manner. Leung et al. [45] provided evidence that ovarian CAFs promote chemoresistance by upregulating lipoma-preferred partner gene in microvascular endothelial cells in a calcium-dependent manner through MFAP5/FAK/ERK signalling. This led to increased vascular leakiness and consequently a reduction in paclitaxel bioavailability. These two examples show clear roles for Ca^2+^ signalling in CAF-mediated chemoresistance, a recognised feature of the TME in tumorigenesis [46,47,48].

Further links between CAFs and Ca^2+^ signalling include colorectal cancer cell secretion of 12(S)-HETE, which promotes CAF migration [49]. Chelation of intracellular calcium reduced 12(S)-HETE-induced CAF migration, demonstrating the requirement of calcium for this migration phenotype in CAFs. Additional support for the role of calcium in CAF activation comes from a high-throughput screen that identified 27 compounds that significantly inhibited CAF activation based on fibronectin expression [50]. Of these 27 compounds, almost a third were glycosides that canonically inhibit Na^+^/K^+^ ATPase and consequently may also alter intracellular calcium stores [51]. Despite these established links between Ca^2+^ signalling and CAFs, there is a lack of calcium signalling studies in CAFs in the context of breast cancer. Additionally, there has been no detailed assessment of potential remodelling of Ca^2+^ influx during CAF activation or the role of Ca^2+^ influx pathways in the conversion of NF to CAFs. To address these gaps, our studies focused on identifying whether breast CAFs display alterations in SOCE in vitro, and if inhibition of specific Ca^2+^ influx pathways can modify the induction of a CAF phenotype.

We optimised CAF induction using TGFβ1 in the human mammary fibroblast cell line (HMF3S). In the literature, TGFβ1-mediated differentiation can take anywhere from 24 h [5,27,52] to three weeks [34]. In agreement with previous studies, we found that 48 h was sufficient for strong induction of a wide range of CAF markers.

SOCE is the most common Ca^2+^ signalling pathway in non-excitatory cells and is altered in phenotypic changes of cancer cells such as EMT [9,49,50]. We hypothesised that SOCE may similarly be altered during CAF activation, and indeed found that SOCE capacity was significantly reduced in TGFβ1-induced HMF3S-CAFs. Interestingly, this was not accompanied by obvious decreases in the expression of primary SOCE components Orai1 and STIM1; instead, Orai1 mRNA levels were slightly elevated. There was, however, a small but significant decrease in levels of STIM2. Remodelling of SOCE could be due to differences in Orai1 plasmalemmal protein levels or trafficking [53,54]. SOCE remodelling is associated with a decrease in VGCC activity in balloon-injured rat aortic cells [20,21,22]. We therefore assessed if this reciprocity may occur in our model. Although there were no pronounced changes in SOCE component gene expression, we did see dramatic upregulation of T- and L-type VGCCs as a consequence of CAF induction. These channels are typically associated with excitable cells like neurons, but also certain CAF phenotypes. For example, L-type VGCCs facilitate PDGF-induced migration of mouse embryonic fibroblasts [55]. This link with migration may partially explain the higher migratory capacity of CAFs [56]. Additionally, CAFs display enhanced contractility [57], and L-type VGCCs are associated with contraction in fibroblasts [55]. CAFs are also more proliferative than NFs and increased T-type calcium current is known to be associated with proliferation in fibroblast-like rat myocytes [58].

To determine if VGCCs are involved in TGFβ1-mediated induction of a CAF-like state in HMF3S breast fibroblasts, we targeted L- and T-type VGCCs with siRNA and found that this reduced the ability of TGFβ1 to upregulate CAF markers. Silencing both channels also reduced αSMA protein expression in the presence of TGFβ1. Similarly, pharmacological inhibition of L- and T-type VGCCs impaired TGFβ1-induced increases in αSMA in HMF3S cells. These studies suggest that L- and T-type VGCCs are regulators of the induction of some CAF markers.

## 5. Conclusions

Using a model of human CAFs, our findings provide new insights into how calcium signalling is remodelled as a consequence of CAF induction. This remodelling appears to be associated with an increase in selected VGCCs, as observed using both our in vitro model and in matched breast cancer patient NF and CAF samples. In our in vitro model, CAF induction was also associated with a functional reduction in SOCE. Pharmacological and siRNA-mediated interference of selected T- and L-type VGCC signalling reduced TGFβ1-induced CAF activation in vitro. The ability of VGCC inhibition to interfere with CAF activation suggests that the selective therapeutic targeting of some VGCCs may attenuate disease progression through reduced induction of NFs into tumour-promoting CAFs.

Future studies should assess the ability of VGCC modulation to regulate other CAF markers and phenotypes such as proliferation, apoptosis resistance, and cytokine release in this model and others, including patient-derived CAFs.

## Figures and Tables

**Figure 1 biomedicines-09-00680-f001:**
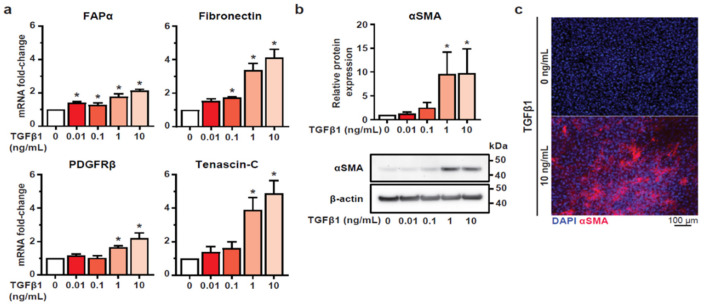
Cancer associated fibroblast (CAF) marker expression in transforming growth factor beta 1 (TGFβ1) treated HMF3S cells. Serum starved HMF3S cells were stimulated with either TGFβ1 (0.01, 0.1, 1, or 10 ng/mL) or vehicle control (0 ng/mL) for 48 h prior to assessing marker. (**a**) TGFβ1 treatment increased mRNA levels of CAF-associated genes fibroblast activation protein-α (FAPα), platelet-derived growth factor receptor-β (PDGFR β), fibronectin and tenascin-C compared with vehicle at the indicated concentrations. (**b**) Representative immunoblot and densitometry showing significant induction of α-smooth muscle actin (αSMA) protein compared with vehicle at 1 and 10 ng/mL TGFβ1. (**c**) Representative immunofluorescence images showing an upregulation of αSMA protein (red) compared with vehicle control. Nuclear (DAPI) staining is also indicated (blue). (**a**,**b**) Data are represented as means ± SD from three independent experiments; ** p <* 0.05 from post-hoc Dunnett’s multiple comparison test after one-way ANOVA revealed significant effect of treatments.

**Figure 2 biomedicines-09-00680-f002:**
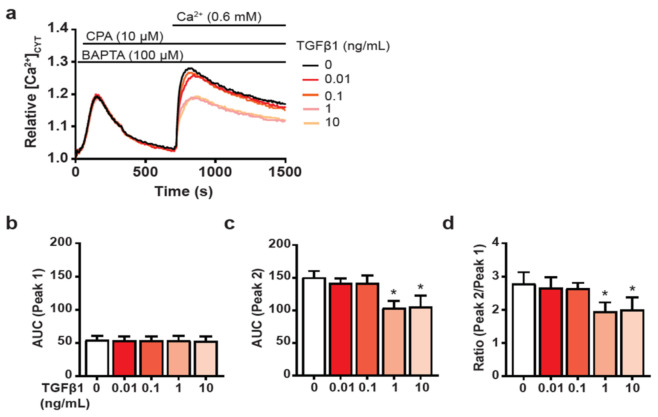
Assessment of cytosolic Ca^2+^ levels during the store-operated calcium entry (SOCE) response in TGFβ1-treated HMF3S cells. (**a**) Serum starved HMF3S cells were treated with TGFβ1 (0.01, 0.1, 1, or 10 ng/mL) or vehicle control (0 ng/mL) for 48 h prior to imaging. To assess potential remodelling of SOCE, cells were sequentially treated with 1,2-bis(o-aminophenoxy)ethane-N,N,N′,N′-tetraacetic acid (BAPTA, 100 μM) to chelate extracellular Ca^2+^, cyclopiazonic acid (CPA, 10 μM) to deplete intracellular Ca^2+^ stores, and Ca^2+^ (0.6 mM) to assess store-operated calcium influx. Calcium traces represent relative cytosolic calcium levels ([Ca^2+^]_CYT_) as mean fluorescence relative to baseline. Quantitation of (**b**) area under the curve (AUC; peak 1), (**c**) AUC (peak 2), and (**d**) ratio AUC (peak 2/peak 1). In (**b**–**d**) data are represented as mean ± SD from three independent experiments. ** p <* 0.05 from post-hoc Dunnett’s multiple comparison test after one-way ANOVA revealed significant effect of treatments.

**Figure 3 biomedicines-09-00680-f003:**
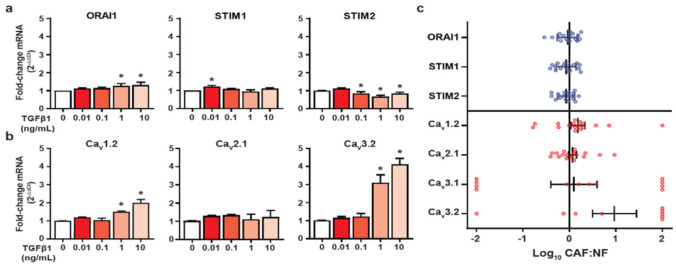
Calcium channels and channel regulators are differentially expressed in CAFs. (**a**,**b**) Serum starved HMF3S cells were stimulated with either TGFβ1 (0.01, 0.1, 1, or 10 ng/mL) or vehicle control (0 ng/mL) for 48 h prior to RNA isolation. mRNA levels of SOCE channels and channel regulators; (**a**) Orai1, stromal interaction molecule 1 and 2 (STIM1, and STIM2) and (**b**) voltage-gated calcium channels (VGCCs); Ca_V_1.2, Ca_V_2.1, Ca_V_3.2, were assessed using quantitative RT-PCR. Values are expressed as fold change relative to vehicle control. Data are represented as mean ± SD from three independent experiments; ** p <* 0.05 from post-hoc Dunnett’s multiple comparison test after one-way ANOVA revealed significant effect of treatments. (**c**) Ratio of mRNA of selected calcium channels and channel regulators in paired patient CAFs and normal breast fibroblasts (NFs). Values ± 2 indicate mRNA was not detected in either the CAF or NF sample.

**Figure 4 biomedicines-09-00680-f004:**
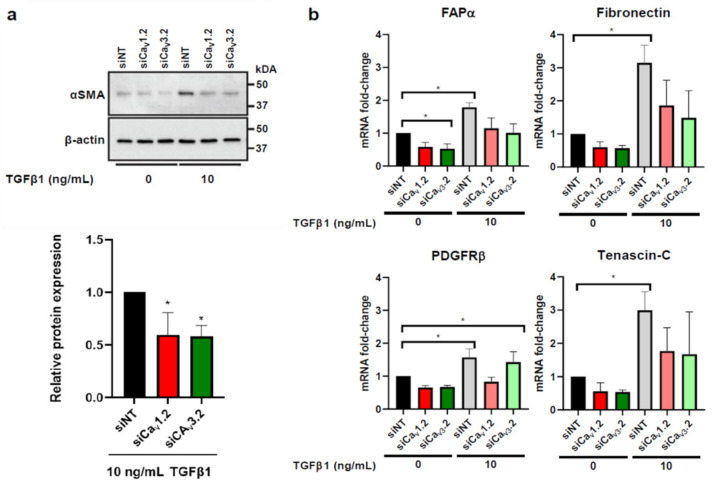
Effect of Ca_V_1.2 and Ca_V_3.2 silencing on TGFβ1-induced CAF marker expression in HMF3S cells. (**a**) Representative immunoblot and densitometry data showing significant downregulation of αSMA protein expression in the presence of siCa_V_1.2 and siCa_V_3.2. (**b**) TGFβ1 treatment increased mRNA levels of CAF-associated genes FAPα, PDGFRβ, fibronectin, and tenascin-C compared with vehicle. siCa_V_1.2 and siCa_V_3.2 reduced induction of FAPα, fibronectin, and tenascin-C while PDGFR-β induction was inhibited with siCa_V_1.2 but not siCa_V_3.2. Data are represented as mean ± SD from three independent experiments; ** p <* 0.05 from post-hoc Dunnett’s multiple comparison test after one-way ANOVA revealed significant effect of treatments.

**Figure 5 biomedicines-09-00680-f005:**
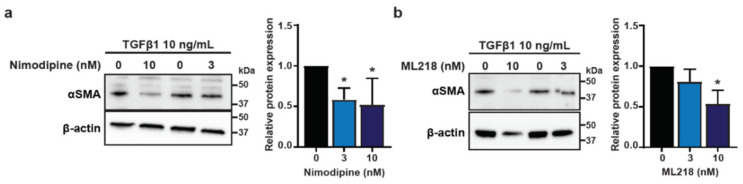
Pharmacological suppression of TGFβ1-induced αSMA expression in HMF3S cells. αSMA protein expression was assessed after treatment with (**a**) nimodipine, and (**b**) ML218. Representative immunoblot and densitometry data showed significant downregulation of αSMA protein with 3 and 10 nM nimodipine and with 10 nM ML218. Data are represented as mean ± SD from four independent experiments; * *p* < 0.05 from post-hoc Dunnett’s multiple comparison test after one-way ANOVA revealed significant effect of treatments.

## Data Availability

The data presented in this study are available in article or Appendix A.

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
