# Peer review of "Altered Calcium Influx Pathways in Cancer-Associated Fibroblasts"

_biomedicines, 2021, doi:10.3390/biomedicines9060680_

Round 1

Reviewer 1 Report

This article describes the research of calcium influx pathways in cancer-associated fibroblasts (CAFs).  They investigated both store operated calcium entry (SOCE) and voltage gated calcium channels (VGCCs). They found two VGCCs, Cav 1.2 and Cav 3.2, significantly impaired CAF activation. In drug discovery, identifying the targets for the disease is the key to it. Their findings are beneficial to understand CAFs. In this article, the studies are very well planned, and the results are clear and easy to follow. I recommend this article to publish without any revision. The results are valuable since the information in this manuscript contains helpful information for further developments in this area.

Author Response

Thank you for the positive feedback.

Reviewer 2 Report

Very good article, clear presentation. A variety of modern techniques applied in the study. I congratulate you for the work. 

The mentioned manuscript has a high level of novelty, illustrated both in the references, but more important, in the selection of materials and methods and of the subject itself (CAFs).  The results are sound, well interpreted and correlated, and also compared to related publications on the topic or closely related topics. So, overall, it is a scientifically sound paper. The overall evaluation of the publication is very good (top 90%), and is OK to publish the manuscript.

Author Response

Thank you for the feedback, based on your request for improvement to the description of the methods we have added additional information to the immunofluorescence methods section detailing the controls and magnification used on page 3 lines 133-135.

We have also expanded the cell culture section outlining the Ca2+ concentration in the media as well as cell confluence on pages 2 and 3 lines 88-91 and 99.

Reviewer 3 Report

Sir, 

I have recently reviewed the manuscript "Altered calcium influx pathways in cancer associated fibroblasts" submitted by Francisco Sadras and co-workers to Biomedicines (MDPI). 

Briefly,  the authors are working on a very relevant topic of the cancer microenvironment - particularly breast cancer. I believe that this is worthy of further research, and it has good translational potential. However, I also believe that there are some outstanding issues in this text which I hope the authors can perfect easily or at least provide honest discussion. 

1) dealing with calcium signalling, I always ask myself what a relevant concentration for experiments in vitro is.  The authors maintain their cell cultures in High Glucose DMEM - as far as my experience says - it is around 1,8mMol Ca++. For some experiments, the authors confirmed this level explicitly. However, this concentration of Ca++ might be a potent prodifferentional signal in, e.g. epithelial cells (namely keratinocytes). Authors should add a brief paragraph on the microenvironmental concentration of calcium and confirm that their model is physiologically relevant. Feel free to start with, e.g. this somewhat older reference (I believe that the subcutaneous concentration mentioned here might be particularly close to your model  - http://currentseparations.com/issues/19-3/19-3d.pdf ) Breast is actually developmentally closely related to the integument. 

2) This issue of epithelial cell differentiation regulated by calcium seems to be also critically important because there are publications dealing with CAF influence on Breast epithelial cell lines differentiation ( go to DOI: 10.1007/S00418-012-0918-3). Is this due to calcium or due to CAF fibroblasts? I hope the authors can briefly comment on this aspect.  Mentioning this, the authors can also offer extrapolation of their observation to other (not breast derived) CAFs elegantly. It is a very important aspect because we have observed similar features in CAFs from skin, breast, pancreas etc...

3) Also, the authors provide a clear protocol regarding cell seeding density (line 95). However, I believe that it is a greatly important issue. The seeding density might be critical in myofibroblasts (aSMA+ fibroblast?) induction. Have you also considered other seeding densities? If so, please, provide details. If not, please comment accordingly. You can start with reference: doi: 10.1073/pnas.93.9.4219

4) Also, the authors mentioned that they use the "HMF3S human breast fibroblast cell line ". However, experiments performed on cell line must also be supported by another one. I believe that Editor could assess this point in a more favourable light because there are some experiments performed on patient-derived CAFs. In my eyes - the core message of the manuscript was confirmed on 17 (seventeen) paired CAF samples from human breast cancer patients, which I find more than satisfactory. 

5) While observing Figure 1,  I must ask myself what the message of ICC image for a novice reader is? Indeed, it leaves a good impression of a remarkably increased quantity of aSMA positive cells (but you present WB, which is more quantitative). However, the presented ICC does not say anything about the cytological features of these cells. Is the aSMA+ cytoskeleton well organised? Or is it just an unstructured cytoplasmic positivity without any pattern? I believe Figure 1 can be greatly enriched by one pair of high-magnification images presenting cytoskeleton (aSMA+) induced by TGFb (btw. authors could here present their negative control, which is missing in and is not described in section 2.4 at all!). 

5) Also, the cell culture timing seems to be surprisingly extremely short (for 48 h stimulation).  I believe that the effective induction of Cancer-associated fibroblast is frequently achieved much later (up to 3 weeks - see: DOI: 10.1159/000324864). I believe this is worthy of comment in the discussion. BTW, this might be confirmed by another cell line - if available. 

6) The authors acknowledged the heterogeneity of fibroblasts in cancer stroma. It is an important aspect and can be easily linked to TGFb signalling as confirmed on single-cell transcriptomic analysis  (DOI: 10.3390/CANCERS12113324). Dealing with this  - it would be interesting to present data on the uniformity of the aSMA induction  (in %) in HMF3S. This might be assessed, e.g. by ICC or FACS. Is there any correlation with the concentration of TGFb used?

Further  - I was also wondering what the authors really mean by using several times the phrase "HMF3S cells following induction of a CAF like phenotype". This is very unlikely that all cells across the population acquire this phenotype uniformly  - see your own ICC data ... Please, reflect this aspect in your discussion. 

7) The authors performed their experiments with pharmacological treatment using, e.g. nimodipine. Is there recently any evidence/indicattion of that nimodipine (or related drugs) could have some beneficial side-effect in cancer patients (suffering from any type of tumour) and thus could be re-purposed in clinical oncology for control of stromal compartment in these malignancies?   

8) Further, there are also available inhibitors of TGFbRI/RI  (e.g. SB 431542 ) hydrate).  Seeing this manuscript from a more distant perspective, impairment of TGFb signalling was linked to some interesting rare cancer hereditary syndromes - like MSSE (Nat Genet. 2011 Feb 27;43(4):365-9. doi: 10.1038/ng.780.) with a high incidence of tumours but surprisingly favourable clinical course leading to regression of epithelial tumours. This is another druggable target relevant to the presented research data...

Minor points

- line 56 :  "...are remodelled in cancer and cancer models" .... please, change the wording. 

- line 90: mycoplasma contamination was carried out every 6 months .... Honestly, it is rather suboptimal frequency. However, the authors can also easily mention that the immunocytochemical staining with DAPI is per se btw. another acceptable way of mycoplasma-free status assessment of their cell cultures. Right? 

TO CONCLUDE: I believe the authors can very easily, in virtually no time, submit a perfected version of their manuscript, and I am very keen to review it again. With all the above-listed modifications/ issues for discussion, I believe this work could be a great contribution to TME  contemporary scientific knowledge and it could be published soon for the benefit of the TME research community. 

Author Response

Thank you for the feedback, please see the attachment for our response. 

Round 2

Reviewer 3 Report

Sir, 

I have reviewed the new version of manuscript "Altered calcium influx pathways in cancer associated fibroblasts" submitted by Francisco Sadras and co-workers. 

Authors provided an extensive rebuttal letter addressing all my previously listed critical comments. I find their answers scientifically sound and honest which I appreciate. 

I do not raise any additional points for discussion and I belive their work is finished. 

The manuscript, as it is submitted now in version 2, is worthy of my full support. I believe that it might be considered as a high-quality work and a great contribution to the scientific knowledge.